# NEUFORM:
# Adaptive Overfitting for Neural Shape Editing

**Connor Z. Lin**
Stanford University

**Niloy J. Mitra**
Adobe / UCL

**Gordon Wetzstein**
Stanford University

**Leonidas Guibas**
Stanford University

**Paul Guerrero**
Adobe

## Abstract

Neural representations are popular for representing shapes, as they can be learned form sensor data and used for data cleanup, model completion, shape editing, and shape synthesis. Current neural representations can be categorized as either overfitting to a single object instance, or representing a collection of objects. However, neither allows accurate editing of neural scene representations: on the one hand, methods that overfit objects achieve highly accurate reconstructions, but do not generalize to unseen object configurations and thus cannot support editing; on the other hand, methods that represent a family of objects with variations do generalize but produce only approximate reconstructions. We propose NEUFORM to combine the advantages of both overfitted and generalizable representations by adaptively using the one most appropriate for each shape region: the overfitted representation where reliable data is available, and the generalizable representation everywhere else. We achieve this with a carefully designed architecture and an approach that blends the network weights of the two representations, avoiding seams and other artifacts. We demonstrate edits that successfully reconfigure parts of human-designed shapes, such as chairs, tables, and lamps, while preserving semantic integrity and the accuracy of an overfitted shape representation. We compare with two state-of-the-art competitors and demonstrate clear improvements in terms of plausibility and fidelity of the resultant edits.

## 1 Introduction

Neural formulations have emerged as an efficient and scalable representation of complex spatial signals, such as radiance fields, 3D occupancy fields, or signed distance functions. These representations are popular as they allow a uniform formulation that can support a range of applications including denoising, data completion, and editing. In the context of shapes, two main types of neural representations have emerged. Starting from an input description (e.g., point clouds, meshes, or distance/occupancy fields), current representations either overfit to a single shape or learn a model that generalizes over a collection of varying shapes. However, neither of the representations alone allows effective shape editing.

Overfitted models [9, 35, 32, 22, 26, 20, 26] reproduce a single shape with high fidelity. While this allows for operations like efficient rendering, surface-based optimization, and data compression, such a representation does not support shape editing or synthesis, since it does not generalize to novel shape configurations.

In contrast, generalizable representations [28, 15, 21, 6] are trained on a large collection of shapes and learn shape priors allowing the representation to adapt to previously unseen shape configurations. Thus, they can be used for shape editing and novel shape synthesis [16, 15, 34, 24, 19, 23]. However, this comes at the cost of a lower-fidelity representation, as the network needs to represent a full dataset and its variations, instead of a single shape. Specifically, these models typically require 'projecting' a

36th Conference on Neural Information Processing Systems (NeurIPS 2022).

shape into the learned latent space before editing it, where the idiosyncrasies of the starting model, in the form of local geometric details, are often lost (see Figure 1).

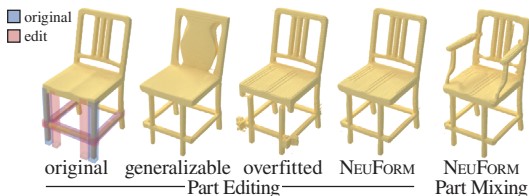

Figure 1: **Adaptive overfitting.** NEUFORM enables detail preserving shape edits that generalize to new part configurations by combining advantages of a generalizable representation (e.g., generation of plausible joint geometry) and an overfitted representation (e.g., detail preservation on the backrest), and also allows mixing parts from different shapes.

We propose a novel blended architecture, called NEUFORM, to combine the advantages of the two representations described above. Specifically, we retain distinctive properties of the input shape by relying on an overfitted model and switch to the generalizable model to complete parts where information is missing (e.g., near new joint locations or regions with holes). The main challenge is to train an adaptive mixing network that blends the information between the overfitted and generalizable models, without introducing artifacts such as undesirable seams or gaps. The NEUFORM architecture allows this seamless sharing of information between the individual networks. Our main technical insights are that (i) it is possible to smoothly interpolate between two neural shape representations by blending between the weights of two networks sharing an architecture and a training history, and (ii) it is possible to do this blending between a generalizable network that works on a global view of the shape and an overfitted network that only has access to part of the shape by carefully pruning the information flow during overfitting.

We evaluate NEUFORM on multiple applications: (i) reconstruction (i.e., projecting a given input to an adaptive overfitted latent space); (ii) part based shape editing; and (iii) shape mixing (i.e., converting an arrangement of parts taken from different models into a coherent shape model). We compare with two state-of-the-art approaches [16, 39] and demonstrate advantages, both quantitatively and qualitatively. Figure 1 shows an example of a shape edit where we can see a clear advantage for NEUFORM over both purely generalizable and purely overfitted representations.

## 2 Related Work

**Single-Scene Neural Shape Representations** Overfitting networks represent one specific shape via a single network by optimizing network weights. Such overfitted networks are useful for several applications including compression [9, 35], adaptive network parameter allocation [20], multiview reconstruction [32, 22], shape optimization [27], or multi-resolution shape representation [26, 38, 35]. While such networks, by construction, accurately capture the original shapes, faithfully encoding their finer details, they can neither be used for editing shapes nor for creating new shapes by combining parts from multiple (source) shapes.

**Multi-Scene Neural Shape Representations** Neural networks have been used to approximate implicit models, as an example of complex spatial functions, to represent shapes as volumetric signed distance fields [28, 6] or occupancy values [21]. Such network learning has been further regularized by geometric constraints like the Eikonal equation [13, 3, 2] or using an intermediate meta-network for faster convergence [18]. Other approaches model shapes using their 2D parameterizations [14, 37]. Improved versions of such methods optimize for low-distortion atlases [4], learn task-specific geometry of 2D domain [10, 31], or force the surface to agree with an implicit function [29]. Most of these methods encode shape collections in a lower-dimensional latent space, as a proxy for the underlying shape space, and support shape editing and generative modeling. For example, sampling from and optimizing in the (restricted) latent spaces can produce voxel grids [19, 12, 5, 8], point clouds [1, 33], meshes [7], or collections of deformable primitives [11]. Others [15, 24, 34] use a two level representations with a primitive-based coarse structure capturing the part arrangement, and a detailing network that adds high-resolution part level geometric details. While these methods do generalize across shapes, and can be used for editing [15, 34, 16, 39], the source models often lose their finer details during the projection to the underlying latent space and subsequent editing process. In Section 4, we compare against two of the most relevant methods: COALESCE [39], which focuses on part-based modeling and synthesizing part connections (i.e., joints), and SPAGHETTI [16], which focuses on inter-part relations towards shape editing and mixing. Our method, NEUFORM, generates higher quality joints than the former while preserving more (original) surface detail than the latter.

## 3 Method

Given a manifold and watertight 3D shape $S$ with known part annotations, our goal is to edit the parts of $S$ without introducing objectionable artifacts or losing geometric detail. The shape can be given as a mesh, signed distance function, or occupancy function, and the part annotations are specified as a set of oriented cuboid bounding boxes $\{C_1, \ldots, C_n\}$, where $n$ is the number of parts of $S$. During editing, parts may be rearranged via scaling and translation, and/or mixed across multiple shapes. To avoid artifacts in the edited shape, some regions of the shape geometry, such as the joints between individual shape parts, need to be adjusted to adapt to the new part configuration. To enable part-based editing without losing geometric detail, we construct two neural representations of shape $S$: a generalizable shape representation and an overfitted shape representation.

The *generalizable shape representation* is a part-aware neural shape representation trained to represent a large shape space. This parameterization can generalize to previously unseen part configurations, including

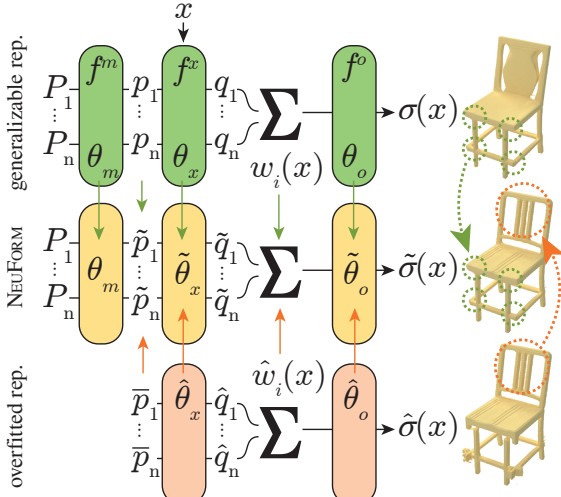

Figure 2: **Architecture overview.** NEUFORM blends between a generalizable neural shape representation (green) and an overfitted neural shape representation (red) by interpolating their network weights and some feature layers. This combines the benefits of detail preservation from the overfitted representation and editability from the generalizable representation.

the edited configuration of shape $S$, but can only provide a low-fidelity reconstruction of $S$.

The *overfitted shape representation* is a neural shape representation overfitted to a single shape $S$. It represents the input shape geometry in great detail, but does not generalize to unseen part configurations, such as edited configurations of $S$.

We combine these representations by blending between them, as explained in Section 3. In regions where reliable data is available for overfitting, such as regions unaffected by edits, we use the overfitted shape representation. In regions where geometry should be adjusted, e.g. joint regions between parts, we leverage the generalizable representation. Both representations share the same architecture and we blend between them by directly interpolating their network parameters, which requires careful design of both the architecture and overfitting setup. We call this approach *adaptive overfitting*.

### Generalizable Shape Representation

**Shape parameters.** In the generalizable representation, a shape $S$ is represented as a set of part parameters $\mathcal{P} := \{P_1, \ldots, P_n\}$. The parameters of a part $P_i := (C_i, g_i)$ consist of a cuboid bounding box $C_i := (v_i, e_i, o_i)$, where $v_i, e_i, o_i$ are the centroid position, size, and orientation of the cuboid, respectively, and a latent vector $g_i$ defining the part's geometry in the local coordinate frame of the cuboid. We obtain $g_i$ from $S$ by encoding $m$ surface and volume points $r_1^i, \ldots, r_m^i$ sampled from part $P_i$ with a PointNet [30] encoder as $g_i := h_\psi(r_1^i, \ldots, r_m^i)$, although other options to obtain $g_i$ such as an auto-decoder setup with inference time optimization are also possible.

**Generalizable occupancy function.** Given the part parameters $P$, a neural network $f_\theta$ models the occupancy field $\sigma_S$ of shape $S$ at any query location $x$ as,

$$\sigma_S(x) \approx \sigma_\mathcal{P}(x) := f_\theta(x|\mathcal{P}). \tag{1}$$

The architecture of $f$ is illustrated in Figure 2. This is similar to the formulation proposed in SPAGHETTI [16], but with changes that are necessary for adaptive overfitting. The network is composed of three parts: A part mixing network $f_{\theta_m}^m$ to exchange information between per-part latent

vectors; a part query network $f_{\theta_x}^x$ to query each part at the query point $x$; and a global occupancy network $f_{\theta_o}^o$ aggregating the results of the per-part queries and output the occupancy at $x$.

*(i) Part mixing network.* The mixing network $f^m$ first converts parameters $P_i$ into per-part latent vectors, and then exchanges information between parts using a self-attention layer:

$$p_i^{\mathcal{P}} := f_{\theta_m}^m(P_i|\mathcal{P}). \tag{2}$$

*(ii) Part query network.* The part query network $f^x$ queries each part $p_i^{\mathcal{P}}$ at the local query point locations using cross-attention from each local query point to all per-part latent vectors $p_i^{\mathcal{P}}$:

$$q_i^{\mathcal{P}}(x) := f_{\theta_x}^x(T_{C_i}^{-1}(x) \mid p_1^{\mathcal{P}} + b_0, \ldots, p_i^{\mathcal{P}} + b_1, \ldots, p_n^{\mathcal{P}} + b_0), \tag{3}$$

where $T_{C_i}^{-1}$ denotes the transformation to the local coordinate frame of $C_i$. Like $f^m$, $f^x$ is run once per part. For a given part $i$, we augment the input latent vectors $p_*^{\mathcal{P}}$ by adding a learned indicator feature that equals $b_1$ for the current part $i$ and $b_0$ for all other parts, giving the network knowledge of which parts it is currently processing. The resulting latent vector $q_i^{\mathcal{P}}$ encodes the local geometry region of part $i$ that is relevant to the query point $x$.

*(iii) Global occupancy network.* Finally, we aggregate the per-part latent vectors $q_i^{\mathcal{P}}$ into a global latent vector using a weighted sum and the global occupancy network $f^o$ computes the occupancy at the query location $x$:

$$\sigma_{\mathcal{P}}(x) := f_{\theta_o}^o\Big(\sum_i w_i^{\mathcal{P}}(x)\, q_i^{\mathcal{P}}(x)\Big), \tag{4}$$

where the weights $w_i^{\mathcal{P}} = \kappa\big(\max(0, d_i^s(x, C_i))\big)$ are based on the signed distance $d_i^s$ from query point to cuboid $C_i$. We choose the triweight kernel for $\kappa$ as it combines a finite support with a smooth falloff: $\kappa(a_i) = (1 - (\frac{a_i}{\rho})^2)^3$, where $\rho$ is the radius of the kernel and $a_i = \min(\max(d_i^s, 0), \rho)$ is the bounded distance to cuboid $C_i$. Essentially, $\rho$ defines the extent of joint regions and $\kappa$ provides a smooth fall-off to 0 as $a_i$ approaches $\rho$. We set $\rho = 0.35$ in all our experiments.

**Training setup.** We jointly train the part encoder $h_\psi$ and the occupancy network $f_\theta$ on a large dataset of shapes $\mathcal{S}$ using a binary cross-entropy loss between the predicted occupancy $\sigma_{\mathcal{P}}(x)$ and the ground truth occupancy $\sigma_S(x)$. More details are given in the supplementary material.

**Shape editing.** Due to training on a large dataset, the generalizable shape representation captures a large space of part configurations. Shape edits can be performed by modifying the parameters of one or multiple cuboids, such as the position $v_i$ or scale $e_i$, to obtain the modified part set $\mathcal{P}_E$ and infer a modified occupancy as $\sigma_{\mathcal{P}_E}(x) := f_\theta(x|\mathcal{P}_E)$.

## Overfitted Shape Representation

**Overfitted occupancy function.** The goal of the overfitted representation is to accurately capture the geometric detail of individual parts of a single shape. We use an overfitted occupancy function $\hat{f}$ with the same architecture as in the generalizable representation to facilitate blending between the two, as described in the next section. Naively overfitting this occupancy function to a shape $S$ would result in artifacts when reconstructing an edited shape $S_E$, since the overfitted occupancy function does not generalize to unseen part configurations. Instead, we carefully sever the information flow between parts during overfitting such that querying the overfitted occupancy function does not use information about the full edited part configuration. We employ a two-part strategy: (i) We freeze the part latent vectors $p_i^{\mathcal{P}}$ before overfitting and only update the query network $f^x$ and the occupancy network $f^o$:

$$\hat{\sigma}_{\mathcal{P}}(x) = \hat{f}_{\hat{\theta}, \overline{\mathcal{P}}}(x|\mathcal{P}) = f_{\hat{\theta}_o}^o\Big(\sum_i \hat{w}_i^{\mathcal{P}}(x)\, \hat{q}_i^{\mathcal{P}, \overline{\mathcal{P}}}(x)\Big), \tag{5}$$

$$\text{with } \hat{q}_i^{\mathcal{P}, \overline{\mathcal{P}}}(x) = f_{\hat{\theta}_x}^x(T_{C_i}^{-1}(x) \mid \overline{p}_1^{\overline{\mathcal{P}}} + b_0, \ldots, \overline{p}_i^{\overline{\mathcal{P}}} + b_1, \ldots, \overline{p}_n^{\overline{\mathcal{P}}} + b_0),$$

where $\hat{\sigma}$ is the occupancy predicted by the overfitted network, $\hat{\theta}_o$, $\hat{\theta}_x$ are the overfitted parameters of the query and occupancy networks, and $\overline{p}^{\overline{\mathcal{P}}}$ denotes part latent vectors that were frozen to the part

set $\overline{\mathcal{P}}$. (ii) We change the weights $\hat{w}_i^{\mathcal{P}}$ to only select the single part latent vector $q^{\mathcal{P}}$ that is closest to the query point $x$: $\hat{w}_i^{\mathcal{P}}(x) = \mathbf{1}_{\{i\}}(\arg\min_i d_i^s(x, C_i))$, where $\mathbf{1}$ is the indicator function. These two changes effectively make the occupancy $\hat{\sigma}_{\mathcal{P}}(x)$ at each query point dependent on only the single closest part, preventing the overfitted occupancy function from being exposed to an unseen part configuration.

**Training setup.** We start with a trained generalizable network $f_\theta$ and a part set $\overline{P}$ we would like to overfit to. We freeze the part latent vectors $\overline{p}_i^{\overline{\mathcal{P}}} = f_{\theta_m}^m(P_i|\overline{\mathcal{P}})$ to the values computed by the generalizable network, and then proceed to overfit both $f_{\theta_x}^x$ and $f_{\theta_o}^o$ to the partset $\overline{\mathcal{P}}$, giving us the overfitted network $\hat{f}_{\hat{\theta}, \overline{\mathcal{P}}}$. During overfitting, we gradually blend between the original weights $w_i$ at the first epoch to the updated weights $\hat{w}_i$ at the last epoch.

**Shape editing.** Similar to the generalizable representation, edits of the overfitted representation can be performed by modifying cuboid parameters to obtain a modified part set $\mathcal{P}_E$, and a modified occupancy $\hat{\sigma}_{\mathcal{P}_E}(x) = f_{\hat{\theta}, \overline{\mathcal{P}}}(x|\mathcal{P}_E)$. As a result of our strategy to decouple parts from each other, a transformation $T_i$ of a cuboid $C_i$ is directly applied to the occupancy of the corresponding part: $\hat{\sigma}_{\mathcal{P}_E}(x) = \hat{\sigma}_{\mathcal{P}}(T_i^{-1}(x))$ for all $x$ that are closer to cuboid $i$ than to any other cuboid. This accurately preserves geometric detail after an edit, but results in discontinuities at the boundaries between edited parts, as shown in Figure 1.

**Adaptive Overfitting** Our goal is to use the overfitted representation in areas where the overfitted occupancy is reliable, and the generalizable representation everywhere else. For shape edits that transform cuboid parameters, the overfitted occupancy in any local region undergoes the same transformation as the nearest cuboid. For human-made shapes such as chairs and tables, this behaviour is desirable in regions that are either close to only one cuboid, or close to only unedited cuboids. In other regions (near joints between two or more cuboids, or where at least one cuboid has been edited), the occupancy may need to undergo more complex transformations to reflect the new part configuration.

Given a set of parts $\mathcal{P}_O$ and an edited version of the parts $\mathcal{P}_E$, we formalize the intuition described above as a scalar blending field $\lambda(x)$ defining a blending factor in $[0, 1]$ between the generalizable and the overfitted representation at each query point $x$:

$$\lambda(x) := \kappa\Big(\min_{C \in (\mathcal{C}^{\mathcal{P}_O} \cup \mathcal{C}^{\mathcal{P}_E}/C_{\min}^{\mathcal{P}_E})} d_i^s(x, C)\Big), \tag{6}$$

where $\mathcal{C}_E^{\mathcal{P}_O}$ and $\mathcal{C}_E^{\mathcal{P}_E}$ are the subsets of cuboids in the original and edited shape, respectively, that have been changed in $\mathcal{P}_E$. $C_{\min}^{\mathcal{P}_E}$ is the cuboid in $\mathcal{P}_E$ closest to $x$. The kernel $\kappa$ is the same triweight kernel defined in Section 3 for part aggregation in the global occupancy network.

Given a blending factor $\lambda(x)$, we finally fuse the two representations by blending between the parameters, weights, and features of the networks:

$$\tilde{\sigma}_{\mathcal{P}}(x) := f_{\tilde{\theta}_o}^o\Big(\sum_i \tilde{w}_i^{\mathcal{P}}(x)\, \tilde{q}_i^{\mathcal{P}, \overline{\mathcal{P}}}(x)\Big), \tag{7}$$

$$\text{with } \tilde{q}_i^{\mathcal{P}, \overline{\mathcal{P}}}(x) = f_{\tilde{\theta}_x}^x(T_{C_i}^{-1}(x) \mid \tilde{p}_1^{\mathcal{P}, \overline{\mathcal{P}}} + b_0, \dots, \tilde{p}_i^{\mathcal{P}, \overline{\mathcal{P}}} + b_1, \dots, \tilde{p}_n^{\mathcal{P}, \overline{\mathcal{P}}} + b_0),$$

where $\tilde{\theta}_o$, $\tilde{\theta}_x$, $\tilde{w}_i^{\mathcal{P}}(x)$, and $\tilde{p}^{\mathcal{P}, \overline{\mathcal{P}}}$ are linearly interpolated between the overfitted and generalizable representation using the blending factor $\lambda(x)$:

$$\tilde{\theta}_* = (1 - \lambda(x))\, \hat{\theta}_* + \lambda(x)\, \theta_*, \tag{8}$$

$$\tilde{w}_i^{\mathcal{P}}(x) = (1 - \lambda(x))\, \hat{w}_i^{\mathcal{P}}(x) + \lambda(x)\, w_i^{\mathcal{P}}(x), \tag{9}$$

$$\tilde{p}_i^{\mathcal{P}, \overline{\mathcal{P}}} = (1 - \lambda(x))\, \overline{p}_i^{\overline{\mathcal{P}}} + \lambda(x)\, p_i^{\mathcal{P}}(x). \tag{10}$$

When editing a shape, we typically overfit to the original configuration of the parts, in that case, we set $\overline{\mathcal{P}} = \mathcal{P}_O$ and $\mathcal{P} = \mathcal{P}_E$.

# 4 Results

We evaluate NEUFORM on three tasks: shape reconstruction, shape editing, and shape part mixing.

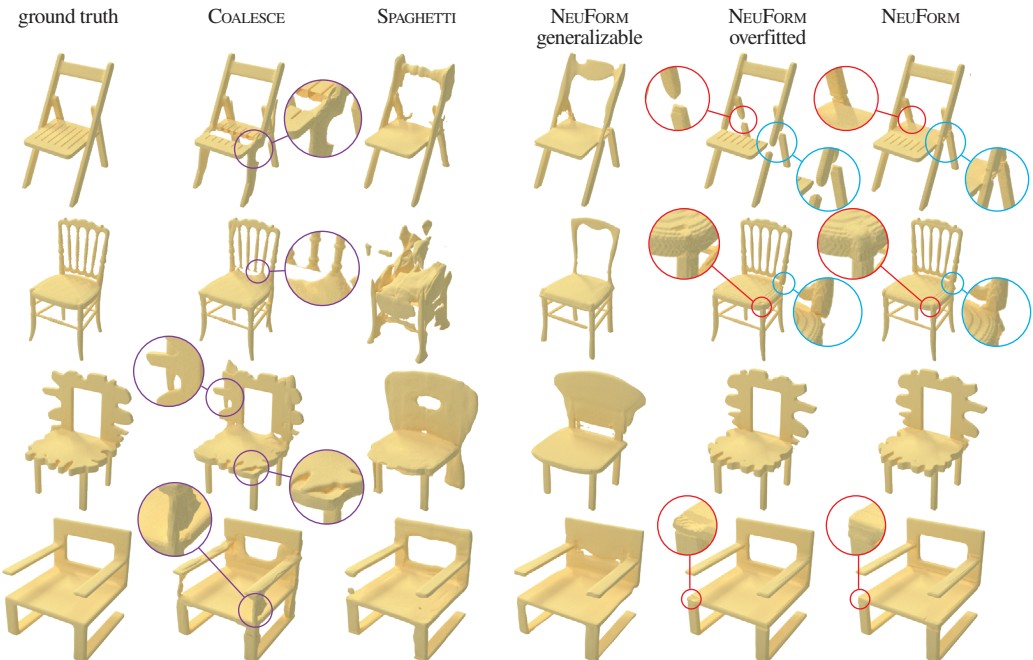

Figure 3: **Shape reconstruction.** Comparing reconstructions of PartNet [25] `chairs`. We show reconstructions of four shapes. COALESCE and the overfitted representation preserve geometric detail, but have more artifacts near joints. SPAGHETTI and the generalizable representation perform better near joints but lose geometric detail. NEUFORM combines the best of both worlds.

**Dataset.** We use the PartNet [25] dataset for our experiments. PartNet is a dataset of human-made shapes in 24 common categories, including furniture and typical household items. Each shape is annotated with hierarchical part segmentation. We experiment on the `chair`, `lamp`, and `table` categories and select hierarchy levels that result in an average of roughly 8, 4, and 8 parts for `chairs`, `lamps`, and `tables`, respectively. Cuboids are computed as oriented bounding boxes of the segmented parts using Trimesh [36]. We train the generalizable model on each shape category separately and choose a training/test split of 6000/1800, 2100/400, and 3500/500 for `chairs`, `lamps`, and `tables`, respectively. All shapes are centered and the largest bounding box side is scaled to 2.

**Training details.** We train the generalizable model for 1000 epochs using the Adam [17] optimizer with a learning rate of $1e-4$ and an exponential learning rate decay of 0.994 per epoch. In each epoch, we train on 4096 query points per shape with a batchsize of 1 shape. We sample 12.5% of the points uniformly in the $[-1, 1]$ cube and 87.5% of the points around the surface with a Guassian offset ($\mathcal{N}(0, 0.05)$). The overfitted model is trained for 100 epochs on a single shape using the same training setup. Training the generalizable model takes roughly 33 hours on a TitanXp GPU and training the overfitted model takes roughly 25 minutes on a single V100 GPU.

**Baselines and ablations.** We compare our results to SPAGHETTI [16] as the state-of-the-art generalizable representation sharing a similar architecture to our generalizable representation, and COALESCE [39], a state-of-the-art method generating the joint geometry between parts given (potentially re-arranged) part meshes. Additionally, we compare with two ablations of our method: using only the generalizable representation and using only the overfitted representation.

**Metrics.** As quantitative metrics, we follow prior work in using the Chamfer Distance (CD) and Earth Mover's Distance (EMD) between points sampled on generated shape surface and points sampled on ground truth shape surfaces. For CD, we sample $30k$ and $10k$ points uniformly on the shape surfaces away from and near joint regions, respectively. We sample 1024 points away from and near joint regions for EMD. As a volumetric measure, we evaluate the signed distance field (SDF) at $25k$ points away from joint regions and $5k$ points near joint regions per shape, with the same distribution as the query points, and report the absolute difference between the values of the generated

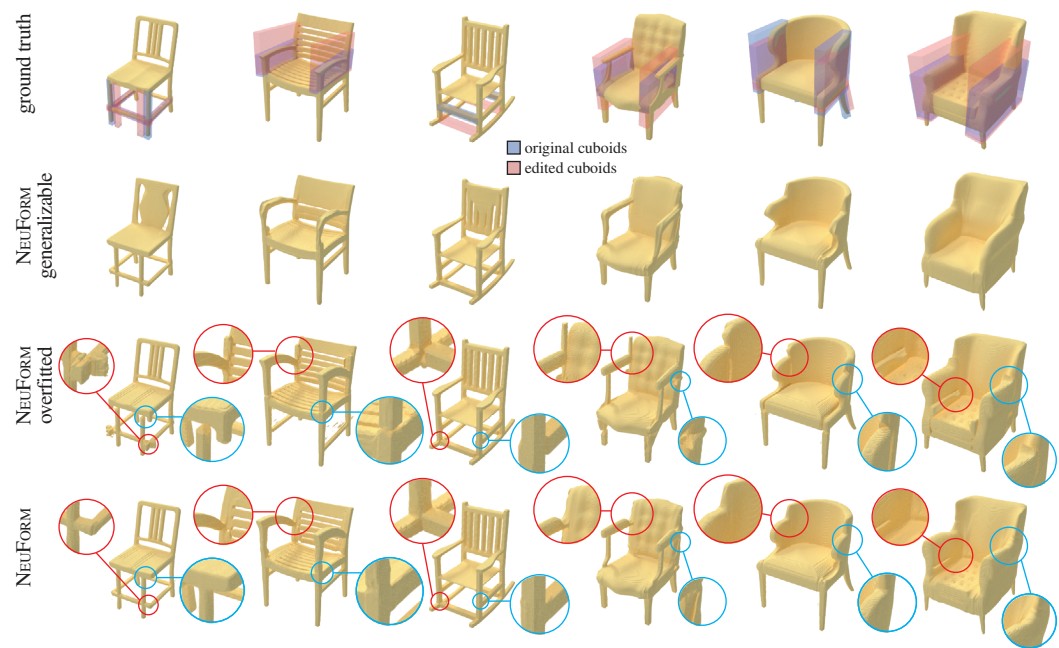

Figure 4: **Shape editing.** Comparing edits on PartNet `chairs` when using only the generalizable or only the overfitted representations. We show edits on shapes with different coarse structure and fine scale details. The generalizable representation has plausible joint areas, but lacks geometric detail; the overfitted representation preserves detail, but has artifacts near joints (see zoom-ins). NEUFORM combines the two representations to both preserve geometric detail and generate plausible joints.

and ground truth shapes. Since our tasks focus on the joints between shape parts, we separately report these metrics on joint regions ($\lambda(x) < 0.5$; see Eq. 6), non-joint regions, and an unweighted average of the two.

**(i) Shape Reconstruction.** While our main application is shape editing, we cannot perform a quantitative evaluation on the shape editing task directly, since we have no ground truth for edited joints. Instead, we use a reconstruction experiment as a pseudo-editing task. We evaluate the reconstruction performance of NEUFORM compared to baselines and ablations on input that reflects the information available during shape editing, where we have some knowledge of non-joint regions, but no knowledge of joint regions. In this experiment, we demonstrate that the generalizable model is necessary to accurately reconstruct/generate joint regions, and the overfitted model is necessary to accurately reconstruct detail in non-joint regions, therefore the comparison includes both overfitted and generalizing models. Our overfitted model is trained without ground truth for any of the joint

Table 1: Comparing shape reconstruction performance. We compare our results to all baselines and ablations. The Chamfer Distance is multiplied by $10^2$. SPAGHETTI and our generalizable representation perform well in joint regions, while COALESCE and the overfitted representation perform better in non-joint regions. The adaptive overfitting performed by NEUFORM achieves good performance in both regions, resulting overall in a significant improvement over both SPAGHETTI and COALESCE. As one would expect, the overfitted representation performs perticularly well on the reconstruction task, but its performance on joint regions drops significantly in shape editing tasks, as we demonstrate qualitatively in the following sections.

| | Joint regions | | | Non-joint regions | | | All regions | | |
|---|---|---|---|---|---|---|---|---|---|
| | CD↓ | EMD↓ | SDF↓ | CD↓ | EMD↓ | SDF↓ | CD↓ | EMD↓ | SDF↓ |
| SPAGHETTI [16] | 0.337 | **65.54** | **1.343** | 1.381 | 176.27 | 3.758 | 0.859 | 120.96 | 2.570 |
| COALESCE [39] | 0.738 | 97.51 | 2.440 | **0.154** | 130.20 | 2.918 | 0.446 | 113.86 | 2.679 |
| NEUFORM generalizable | 0.390 | 84.27 | 2.109 | 0.523 | 117.81 | 5.208 | 0.457 | 101.04 | 3.659 |
| NEUFORM overfitted | 0.318 | 78.54 | 2.198 | 0.157 | **80.45** | 2.644 | **0.238** | **79.50** | 2.471 |
| NEUFORM | **0.253** | 78.05 | 1.814 | 0.334 | 88.53 | **2.538** | 0.293 | 83.29 | **2.176** |

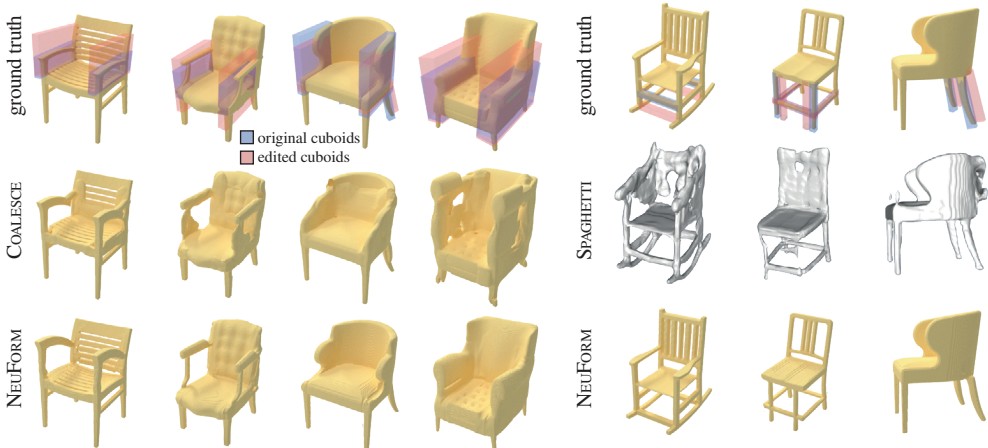

Figure 5: Comparing edits on PartNet `chairs` to COALESCE [39] and SPAGHETTI [16]. We show two different sets of edits because COALESCE does not support edits of more fine-grained parts like bars, while SPAGHETTI does not currently support part scaling in their released code. COALESCE struggles with more extended joint areas and SPAGHETTI's result is significantly noisier after an edit. Here we show screenshots from SPAGHETTI's editing UI (hence the different color). Blending between the generalizable and overfitted representations using NEUFORM gives us more plausible edit results, with cleaner joints and detailed part geometry.

areas and we perform the comparison on 64 shapes selected randomly from the test set. COALESCE does not support fine-grained parts, thus, for a fair comparison, we restrict our joint areas to those defined by COALESCE in this experiment.

Table 1 shows quantitative results of this comparison and Figure 3 shows qualitative examples for all methods. SPAGHETTI performs well in joint regions, but since it is a generalizable model, it lags behind the overfitted model and COALESCE in non-joint regions, giving a lower performance overall. COALESCE has the lowest performance in joint regions, as it struggles with larger or more extended joint areas, and has reasonable performance in non-joint areas. While COALESCE uses the ground truth geometry in non-joint areas, some of the joint geometry tends to incorrectly extend into the non-joint areas, lowering the performance. As expected, our generalizable representation performs well in joint regions, and misses detail in non-joint regions. In this reconstruction task, the overfitted representation performs significantly better in joint regions than in the edit tasks we describe in the next sections, since the part configuration of the reconstructed shape is the same as the part configuration it was overfitted to. In the reconstructions, errors at the joints are due to the missing ground truth in joint regions. NEUFORM combines the advantages of the overfitted- and the generalizable representations. It uses the lower-detail, but plausible geometry from the generalizable model in the joint regions, and naturally transitions to the detailed geometry of the

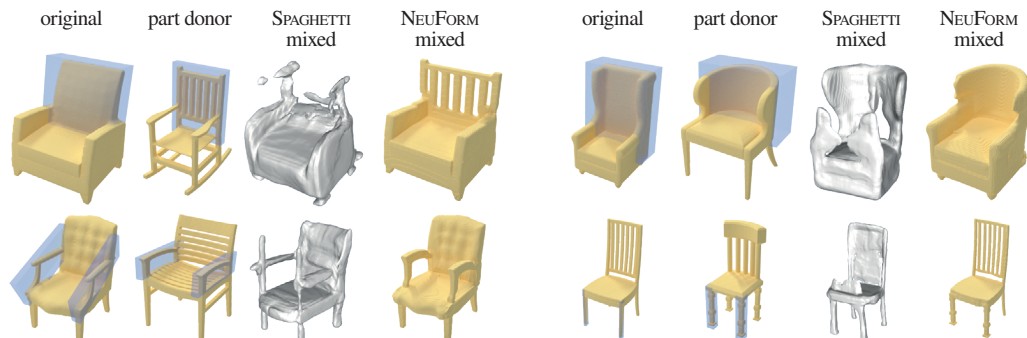

Figure 6: **Shape mixing.** Mixing parts of different PartNet `chairs`. We replace the highlighted part in the original shape with the highlighted part in the donor shape, and compare our results to SPAGHETTI on re-mixed shapes. Similar to the editing setting, SPAGHETTI's quality deteriorates on shapes with mixed parts. NEUFORM combines the foreign part more seamlessly into the shape.

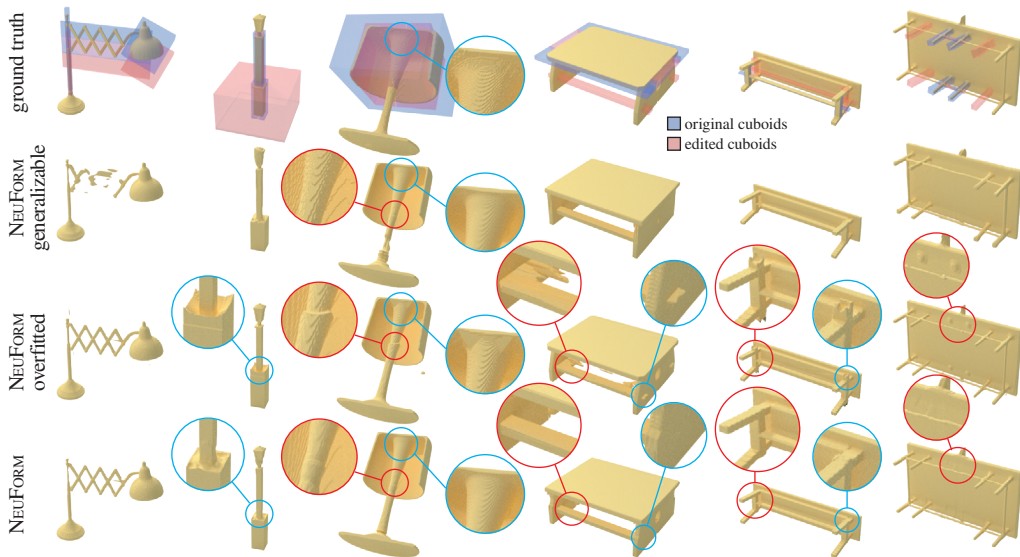

Figure 7: **Different object categories.** Shape edits on PartNet `tables` and `lamps`. Similar to `chairs`, the generalizable model lacks detail and the overfitted model contains artifacts in joint regions, whereas NEUFORM combines the advantages of both.

overfitted representation in non-joint regions. Note that the exact extent of the joint regions is defined by the blending field $\lambda$ defined in Eq. 6.

**(ii) Shape Editing.** We experiment with shape edits by modifying the parameters of one or multiple cuboids of our shape representation. Editing results of NEUFORM compared to the generalizable and overfitted representations are shown in Figure 4. Edits on the generalizable representation confirm the trend we saw in the reconstruction task: joints are plausible after edits, but geometric detail is not preserved. When editing the overfitted representation, we observe significant artifacts near the joints, due to the previously unseen part configuration. Our adaptive overfitting strategy preserves the plausible joints of the generalizable representation as well as the geometric detail of the overfitted representation.

In Figure 5, we compare shape editing to COALESCE and SPAGHETTI. Since COALESCE does not supports fine-grained edits, and SPAGHETTI does not support scaling, we compare to each on a separate set of edits. As we saw in the reconstruction, COALESCE struggles with extended joints, while SPAGHETTI's geometry deteriorates significantly after an edit.

**(iii) Shape Mixing.** We demonstrate our model's ability to assemble new shapes from the parts of pre-existing ones in Figure 6. We mix and match cuboids and their associated part features from different `chairs`, and then blend the parts together. For a given query point and its closest part $P$, we use the overfitted representation associated with the shape that $P$ was originally part of. Our method synthesizes much smoother joint connections between parts while preserving their surface details.

**Additional shape categories.** Figure 7 shows edit results on `tables` and `lamps`, compared to the generalizable and overfitted representations. Similar to `chairs`, the generalizable representation is missing shape detail, resulting, for example, in artifacts on thin parts, while the overfitted representation struggles with joint areas. In the right-most `table`, we can clearly see that these artifacts occur both in regions that are joints after the edit, as well as regions that used to be joints in the original shape. Adaptive overfitting avoids these artifacts.

**Overfitting times.** We evaluate the time required for training the overfitted model by measuring reconstruction quality when overfitting for different amounts of time. Times were measured on a single V100 GPU. Results are shown in Figure 8. We can see that most regions are sufficiently converged after 8 minutes, showing that the 25 minutes used for the main experiments is chosen very conservatively and may be reduced without losing significant quality.

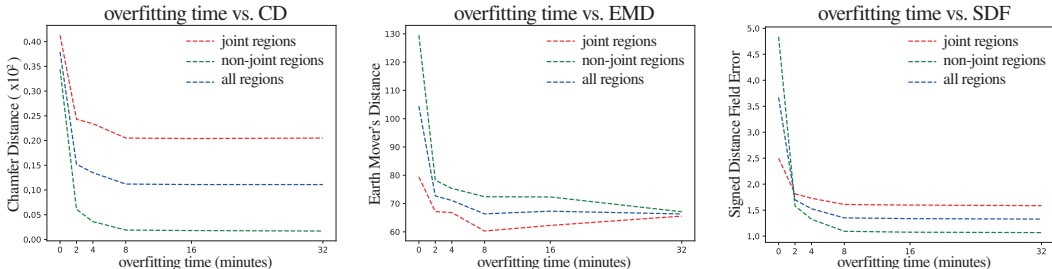

Figure 8: **Overfitting time vs. reconstruction quality.** We show the reconstruction quality at different overfitting times, where we measure reconstruction quality in terms of CD, EMD, and SDF error. Note how quality typically converges around 8 minutes.

## 5 Conclusions

We have introduced the NEUFORM architecture to enable adaptive mixing of information between a generalizable neural neural network, trained on a collection of shapes, and an overfitted model, trained on a single shape to capture its idiosyncrasies. We achieved this by designing a network architecture that allows adaptive mixing of networks by carefully blending respective network weights and training history.

Our work is just the first step in the direction of merging overfitted and generalizable models. For example, currently the two models do not have explicit knowledge of each other, adding this knowledge could be interesting future work. For shape editing, this could allow the generalizable network to focus more on joint geometry. Another limitation is the currently non-data-driven blending field. Learning a context-based blending factor is a promising next step for facilitating easier and higher quality editing. Shape compression may be a possible future application, where a generalizable prior could efficiently represent the coarse geometry of a shape, and an overfitted model could be trained to represent details on top of the generalizable prior.

## Acknowledgments and Disclosure of Funding

This work was supported by a David Cheriton Stanford Graduate Fellowship, ARL grant W911NF-21-2-0104, a Vannevar Bush Faculty Fellowship, a PECASE by the ARO, Stanford HAI, and gifts from the Adobe and Snap Corporations.

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
