# NEUFORM:
# Adaptive Overfitting for Neural Shape Editing Supplementary Material

**Connor Z. Lin**
Stanford University

**Niloy J. Mitra**
Adobe / UCL

**Gordon Wetzstein**
Stanford University

**Leonidas Guibas**
Stanford University

**Paul Guerrero**
Adobe

## S1   Overview

In this supplementary document, we provide additional details on our data preparation procedure (Section S2), our architecture (Section S3), and the baselines (Section S4). Additionally, we provide a quantitative evaluation of the shape editing experiments (Section S5), extend the shape mixing experiments to include a comparison to COALESCE (Section S6), and provide several additional ablations of our approach (Section S7).

## S2   Data Preparation

We use the PartNet [3] dataset for training and evaluation. The PartNet dataset defines a hierarchical decomposition of each shape into parts. We use the first level of the part hierarchy except for the base of chairs and tables, where we use a deeper level to obtain individual legs and bars. To compute ground truth occupancy, we make the shapes watertight using an existing method [2]. Next, we center each shape at the origin and scale it such that the largest extent along any axis is $[-1, 1]$. We fit oriented boundary boxes to parts using TriMesh [5]. To obtain a point cloud for the part geometry encoder $h_\psi$, we uniformly sample 5k surface and volume points for each part, additionally storing the SDF gradient for each point (the SDF gradient generalizes the surface normal to the volume), and transforming the resulting point cloud into the local coordinate frame of the part's cuboid. Finally, to obtain the query points $x$ used during training, we sample 50k points uniformly inside the bounding cube $[-1, 1]^3$ and 50k points on the surface of the mesh with added Gaussian noise of $\sigma = 0.05$.

## S3   Architecture Details

**Part Encoder $h$.**   We encode the geometry of each part using a PointNet [4] encoder consisting of six hidden layers prior to the Max-Pooling layer and two hidden layers after pooling. The hidden layers start with dimensionality of 64 and consecutively double until reaching dimensionality 512. As input, we randomly sample a subset of 4096 surface and volume points for each part, taken from the point cloud we pre-computed during the data preparation step (see Section S2). We input both the point locations and SDF gradients, resulting in a 6-dimensional input vector per point. The output is a 512-dimensional feature vector $g_i$ per part that captures the part geometry in the local coordinate frame of the part's cuboid.

**Part mixing network $f^m$.**   The part mixing network performs two main operations: it first combines the geometry feature vector $g_i$ and the cuboid parameters $C_i$ of each part into a per-part feature vector $p'_i$, and then exchanges information between parts using a self-attention layer to obtain an updated per-part feature vector $p_i^{\mathcal{P}}$. To obtain the per-part feature vector $p'_i$, a 512-dimensional cuboid feature vector $c_i$ is computed from the cuboid parameters $C_i$ using a three-layer MLP with 512 hidden

36th Conference on Neural Information Processing Systems (NeurIPS 2022).

dimensions, and then added to the feature vector $g_i$: $p_i' = g_i + c_i$. Similar to SPAGHETTI [1], we use multiple self-attention layers to mix information between the per-part feature vectors $p_i'$ to obtain updated feature vectors $p_i^{\mathcal{P}}$:

$$\{p_i^{\mathcal{P}}\}_i = \texttt{SAtt}^4(\{p_i'\}_i) \tag{S1}$$

where $\texttt{SAtt}^4$ denotes four Transformer [6] self-attention blocks. Each block includes an attention layer with 8 attention heads, followed by a feed-forward layer. See the original Transformers paper [6] for details.

**Part query network $f^x$.** The part query network queries all parts at a query point $x$ by performing cross-attention from the query point to all parts. Since the part geometry is defined in local coordinates of the part's cuboid, we transform the query point into local coordinates $x_i^l = T_{C_i}^{-1}(x)$, where $T_{C_i}^{-1}$ is the transformation to the local coordinate frame of the cuboid $C_i$. We use a learned positional encoding $\pi$ for the local coordinates $x_i^l$, and perform cross-attention from each encoded local coordinate to all cuboids:

$$\{q_i^{\mathcal{P}}(x)\} = \texttt{CAtt}^4(\{\pi(x_i^l)\}, \{p_1^{\mathcal{P}} + b_0, \ldots, p_i^{\mathcal{P}} + b_1, \ldots, p_n^{\mathcal{P}} + b_0\}), \tag{S2}$$

where $\texttt{CAtt}^4(a, b)$ denotes four Transformer cross-attention blocks, with queries based on $a$ and keys/values based on $b$. Each block includes an attention layer with 8 attention heads followed by a feed-forward layer. See the original Transformers paper [6] for details. Note that a different set of keys/values is used for each query, since the indicator feature $b_1$ is added to a different part feature vector for each local query point: for query point $x_i^l$, it is added to the part feature vector $p_i^{\mathcal{P}}$.

**Global occupancy network $f^o$.** The global occupancy network is implemented as a two-layer MLP with 512 hidden dimensions.

## S4   Baseline Details

**COALESCE [7].** We use the pre-trained model provided by the authors and pre-process all shapes using the approach described in COALESCE, making sure to re-normalize the shapes so the scaling and orientation is comparable to the existing test set shapes. Since our cuboids use a more fine-grained shape decomposition than COALESCE, we assign each of our cuboids to one of the shape parts defined by COALESCE and treat each resulting group of cuboids as a single part. We then define a segmentation of the shape by assigning each surface point to the cuboid it has the smallest signed distance to and remove the surface within a small radius of segment boundaries, as described in the COALESCE paper. The output of COALESCE is transformed back to our normalized coordinates for comparison with the ground truth.

**SPAGHETTI [1].** Here, we also use the pre-trained model provided by the authors and make sure to normalize the shapes as required by SPAGHETTI. Unlike COALESCE, we can work directly with our cuboids, as SPAGHETTI can handle fine-grained parts and shares our cuboid representation. When editing or mixing shapes, we use the editing UI provided by the authors (we do not need to use the UI for shape reconstruction). The output of SPAGHETTI is transformed back to our normalized coordinates for comparison with the ground truth.

## S5   Quantitative Evaluation of Shape Edits

In this section, we show a quantitative evaluation of the edits shown in Figure 4 of the main paper. Since we do not have ground truth for a shape with an edited cuboid configuration, we do the inverse: we start with the edited cuboid configuration and overfit to it (i.e. $(\overline{\mathcal{P}}, \mathcal{P}) = (\mathcal{P}_E, \mathcal{P}_O)$ instead of $(\overline{\mathcal{P}}, \mathcal{P}) = (\mathcal{P}_O, \mathcal{P}_E)$ as described at the end of Section 3 in the main paper). Then, we re-arrange the edited cuboids to undo the edit. Since this should result in the original shape, we do have ground truth for this re-arrangement that we can use to compute the quantitative metrics defined in Section 4 of the main paper. Note that it is possible to overfit to the edited cuboid configuration, since our overfitted model only requires ground truth in non-joint regions for training, which we can obtain by simply transforming individual parts geometries.

Results are shown in Table S1. We can see that the generalizable model performs better than the overfitted model in joint regions, while the reverse is true for non-joint regions. NEUFORM combines

Table S1: **Quantitative comparison of chair edits.** We show a quantitative evaluation of the edits shown in Figure 4 of the main paper. The generalizable model performs better than the overfitted model in joint regions, while the reverse is true for non-joint regions. NEUFORM combines the advantages of both an performs better on average in all regions.

| | Edited joint regions | | | Non-joint regions | | | All regions | | |
|---|---|---|---|---|---|---|---|---|---|
| | CD↓ | EMD↓ | SDF↓ | CD↓ | EMD↓ | SDF↓ | CD↓ | EMD↓ | SDF↓ |
| NEUFORM generalizable | 0.052 | **48.07** | 0.745 | 0.042 | 72.31 | 1.751 | 0.047 | 60.19 | 1.248 |
| NEUFORM overfitted | 0.110 | 64.89 | 1.159 | 0.020 | 69.12 | 0.687 | 0.065 | 67.00 | 0.923 |
| NEUFORM | **0.052** | 49.72 | **0.642** | **0.019** | **62.66** | **0.684** | **0.036** | **56.19** | **0.663** |

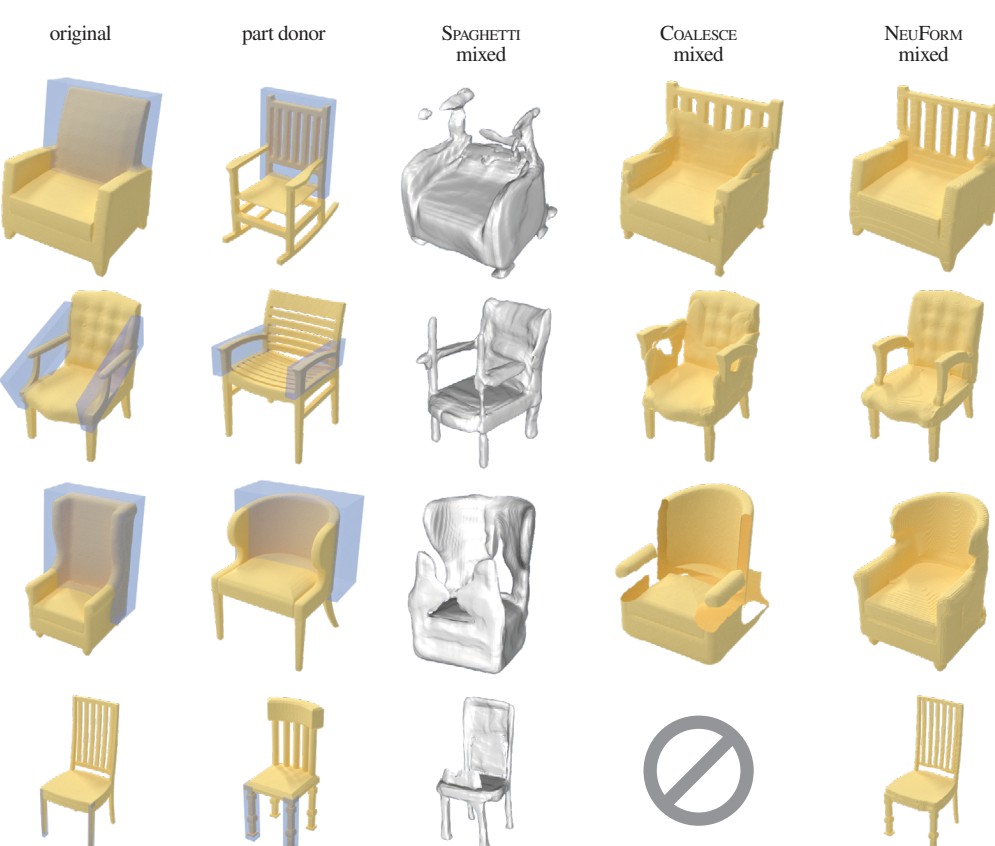

Figure S1: **Shape mixing with COALESCE.** We extend the shape mixing results shown in Figure 6 of the main paper by adding a comparison to COALESCE. Joints produced by COALESCE are generally more noisy. In row three, one of the steps in the pipeline of COALESCE fails, producing no joint geometry and the edit in row four COALESCE is not applicable, since fine-grained part edits like individual chair legs are not supported. NEUFORM produces more plausible results with fewer artifacts.

the advantages of both an performs better on average in all regions. Note that NEUFORM even slightly outperforms the generalizable model in joint regions and the overfitted model in non-joint regions, since both the joint regions and the non-joint regions include small transition regions between joints and non-joints that NEUFORM performs better on than either overfitted or generalizable model alone.

## S6  Part Mixing Comparison to COALESCE

In Figure S1, we extend the part mixing experiments shown in the main paper with a comparison to COALESCE [7]. Similar to the editing results in Figure 5 of the main paper, we can see that COALESCE preserves geometric detail of individual parts. But as the COALESCE authors note in their limitations, the method struggles to connect parts with stronger geometric or topological

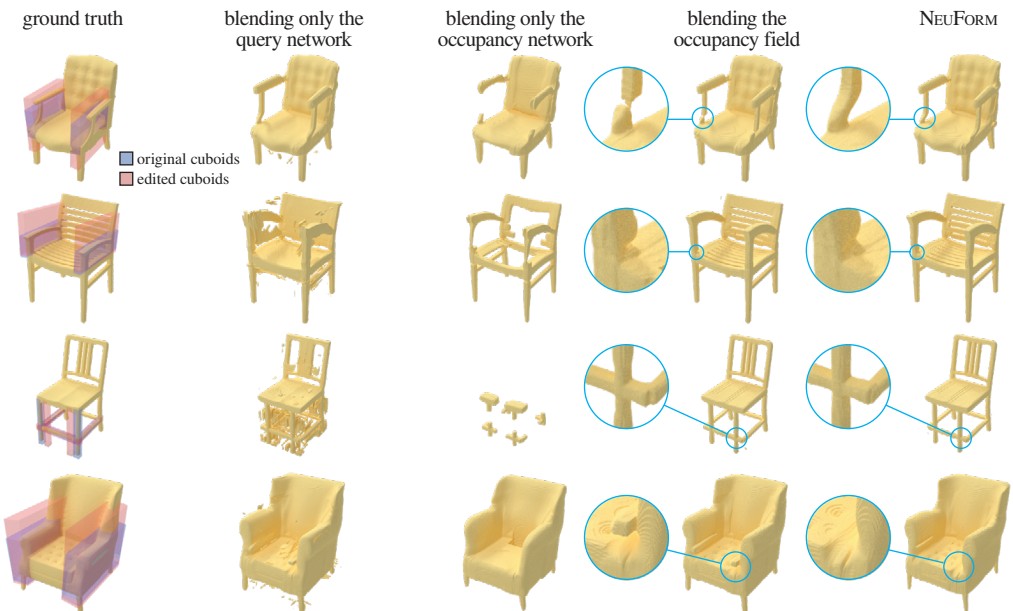

Figure S2: **Additional ablations.** We show ablations that only blend with the overfitted network for a subset of the networks (second column: only the part query network $f^x$, third column: only the global occupancy network $f^o$), and an ablation that directly blends the occupancy fields (fourth column). Blending only a subset of networks results in severe artifacts, while directly blending the occupancy fields gives overall better results, but shows artifacts in joints where there is larger disagreement between the generalizable and the overfitted representations.

incompatibility, resulting in noisy joints four our shape mixing examples. In the example in row three, the Poisson blending step of COALESCE fails, completely removing any joint geometry. In row four, we can see another limitation of COALESCE that the authors point out in their paper: editing or mixing fine-grained parts like individual chair legs is not supported, due to the larger inconsistency between the part decompositions of different chairs in the dataset when using more fine-grained parts.

## S7  Additional Ablations

We show three additional ablations qualitatively in Figure S2. First we show the effect of only blending a subset of our networks instead of blending both $f^x$ and $f^o$. Results are shown in the second and third columns. Since this results in a parameter combinations that were not seen during training, results show severe artifacts. Next, we show a possible alternative to blending the overfitted and generalizable representations in network parameter space: we show directly blending the occupancy fields output by the two representations. This seems to work well at first glance, but on closer inspection, we can see that it results in artifacts in regions with larger disagreement between the overfitted and the generalizable representations. For example, this is clearly visible in the region highlighted in Figure S2, first row, fourth column. The results of NEUFORM show that blending in the network parameter space handles disagreement between the overfitted and the generalizable representations more gracefully.