# OpenReview forum: "NeuForm: Adaptive Overfitting for Neural Shape Editing"
_NeurIPS.cc/2022/Conference — NeurIPS 2022 Accept_

### Official Review · Reviewer_iV53 · 2022-07-11

**Rating:** 7
**Confidence:** 5
**Soundness:** 4 excellent
**Presentation:** 3 good
**Contribution:** 3 good

**Summary:**

This paper presents a shape editing method which works by combining generalized and overfit occupancy models. By applying the generalized model to edited regions, the overfit model to unedited regions and smoothly interpolating between them they naturally achieve more attractive shape predictions. With this method they demonstrate superior shape reconstructions, shape edits and and shape mixing.

**Questions:**

Why is the overfit model trained without joint ground truth in the reconstruction task? I am not sure it makes sense to evaluate the overfit  or the NEUFORM models in this setting as reconstruction is a generalization task and you are overfitting to shape in both situations. I may be miss-understanding something here though.

**Limitations:**

Yes , though I mention above a possible addition.

**Strengths And Weaknesses:**

The strengths of this paper is that the idea is presented and works well. I have no qualms with the methodology, presentation or results.

My main concern is the time scale needed to begin editing a new shape. It is indicated that it takes 25 mins to train an overfitting model, which if I understand will be required for each new shape you wish to edit. This seems prohibitively long and I think should be addressed in the paper somewhere, with a table comparing the time to edit for the various methods you compare to. At the very least it should be mentioned in the limitations.

With that said the editing and mixing results are impressive, and the idea and results are presented very well.

---

> ### Author Response · Authors · 2022-08-01
> **Rebuttal**
>
> Thanks for the feedback and suggestions. We will correct all smaller issues; below we address the main concerns:
>
> ### Long overfitting times
> We used a conservatively long overfitting schedule that was shared by all shapes in our experiment, to make sure the overfitting results for each shape had fully converged. Note that this overfitting already starts from the generalizable model and refines it. We can reduce this time without significant loss in accuracy, for example overfitting for only 5 minutes increases Chamfer distance and SDF error by only ~10% (0.017 to 0.019 CD and 0.679 to 0.744 SDF - these numbers are roughly comparable to Table S1 in the supplementary, but not fully comparable, since they were computed on a smaller subset than Table S1). Additionally, there has been active research aimed at reducing the overfitting time for neural representations (mostly for NeRFs), for example using hash tables and CUDA kernels [A] which takes order of a minute. Adapting this work to our setting seems like a fertile direction for future research that we hope can draw inspiration from our method.
>
> ### Justification for the experimental setup of the reconstruction experiment
> Indeed, having ground truth for shape editing is tricky. We cannot perform a quantitative evaluation on the shape editing task directly, since we have no ground truth for edited joints. Instead, we use the reconstruction experiment as pseudo-editing task to perform a quantitative comparison on input that reflects the information available in our main application: shape editing, where we have some knowledge of non-joint regions, but no knowledge of joint regions. In this experiment, we want to demonstrate that the generalizable model is necessary to accurately reconstruct/generate joint regions, and the overfitted model is necessary to accurately reconstruct detail in non-joint regions, therefore the comparison includes both overfitted and generalizing models. We will clarify this when describing the experimental setup.
>
> ### Reference
> [A] Instant Neural Graphics Primitives with a Multiresolution Hash Encoding, Müller et al., Siggraph 2022

---

### Official Review · Reviewer_WuvA · 2022-07-12

**Rating:** 8
**Confidence:** 4
**Soundness:** 4 excellent
**Presentation:** 4 excellent
**Contribution:** 4 excellent

**Summary:**

The paper presents NEUFORM, a neural representation that combines the benefits of detail preservation of overfitted representations and edit-ability of generalizable representations. In the generalizable representation the shape is represented as a set of part parameters and an occupancy function models the occupancy field of the shape at any query location. The model consists of a part mixing network, a part query network, and a global occupancy network. An overfitted occupancy function with the same architecture as the generalizable representation is used to capture geometric details. The part latent vectors are frozen before overfitting and only the query network and the occupancy network are updated. A scalar blending field defines a blending factor between the generalizable and overfitted representations at each query point. The two representations are fused by blending between the parameters, weights and the features of the networks given the blending factor. NEUFORM is evaluated on reconstruction, part based shape editing and shape mixing. Qualitative and quantitative experiments demonstrate superiority of NEUFORM over SPAGHETTI [17] and COALESCE [39] as the baselines.


**Questions:**

- As commonly done in prior work, the authors train separate models on each category, and they do not discuss whether training a generalizable model jointly on multiple categories help or hurt the performance. The model architecture is not directly tied to a fixed category, and shapes from different categories can share similar-looking parts. Is it possible to share knowledge across categories to further improve reconstruction and editing?
- It would be good to have the same surface color for shapes from NEUFORM vs SPAGHETTI in Figures 5 and 6.

**Limitations:**

- There is no potential negative societal impact.
- The authors note limitations of their work in terms of limited explicit knowledge-sharing between over-fitted and generalizable models and non-data-driven blending field.

**Strengths And Weaknesses:**

**Originality**:
- The proposed idea of combining overfitted and generalizable representation by adaptively blending corresponding network weights is novel and interesting.
- The authors notice that the problem of shape editing needs both detail preserving and generalizable representations, and it lends itself well to the proposed adaptive model.
- The authors carefully design the generalizable and overfitted networks so that they can be combined well in a hybrid model.
- They use the idea of blending network weights which is trending topic in machine learning but less studied in 3D deep learning.

**Quality**:
- The paper is well-motivated and well-written.
- Experiments are thorough and convincing and sufficient details are provided for the experiments to be reproducible.

**Clarity**:
- The proposed approach is described clearly. The authors provide intuitive explanation of their choices as well as mathematical abstraction of the proposed model.

**Significance**:
- The problem of shape reconstruction and editing is of great practical value and the current state-of-art methods create noticeable artifacts and are far from perfect.
- The paper provides significant value to the research community as it shows considerable quantitative and qualitative improvements over the state-of-the-art across tasks such as reconstruction, shape editing and shape mixing.

---

> ### Author Response · Authors · 2022-08-01
> **Rebuttal**
>
> Thanks for the feedback and suggestions. We will correct all smaller issues; below we address the main concerns:
>
> ### Training on multiple shape categories
> This is an interesting direction to explore. On one hand, it would increase the diversity of edits that can be handled, since the generalizable model would be exposed to a larger range of part configurations. On the other hand, it would likely reduce the fidelity for individual shapes, or would require a larger network capacity. Some shape categories such as tables and chairs also have a large overlap in their configuration and geometry spaces, making joint training on these categories easier. We will add a discussion to our future work passage.
>
> ### Same color for SPAGHETTI results in Figures 5 and 6
> We understand the concern, but unfortunately shape editing is only available via an interactive UI in SPAGHETTI. Exporting shapes from this UI would require significant code changes. We would coordinate with the SPAGHETTI authors to implement exporting shapes from the UI should our paper get accepted.

---

### Official Review · Reviewer_JEir · 2022-07-15

**Rating:** 7
**Confidence:** 4
**Soundness:** 3 good
**Presentation:** 4 excellent
**Contribution:** 3 good

**Summary:**

This paper proposes an architecture and training scheme for 3D shape reconstruction that combines the generalizability of models that parameterize an entire shape family with the level of detail of models overfit to a single shape. To this end, a generalizable part-based model is trained for the shape family and another model with the same architecture is overfit to a specific shape. The two models are then combined in order to take advantage of edits via the latent space of the former while keeping the details captured by the latter. As a result, it is possible to perform part-based edits where novel geometry is hallucinated where necessary (i.e., around part joints) while existing geometry (the parts themselves) are faithfully reconstructed.

The authors show experimental results on sheep reconstruction, shape editing, and shape mixing, and they compare to state-of-the-art part-based deep shape reconstruction methods.

**Questions:**

Looking at certain results, such as the first row of Figure 3, it seems that the model is sometimes unable to extrapolate local part texture/style when parts are edited (e.g., the slate on the chair end halfway up the seat on the edited model).

Can this method be leveraged for compression in a superior way than previous similar models? Is there any way to significantly shrink the generalizable model, since much of the burden of representation is now carried by the overfit model?

Is it possible to improve the time required to adapt this approach to every new 3D model (e.g., retraining the overfit model)? This seems to be a bottleneck compared to using a purely generalizable model; could some initialization or fine-tuning alleviate this?

Minor typos:

L2: "form" -> "from"

L139: $P$ -> $\mathcal P$

L188: $\overline P$ -> $\overline {\mathcal P}$

L232: $1e-4$ -> "1e-4" or "0.0001"

**Limitations:**

Limitations and directions for future work are adequately discussed.

**Strengths And Weaknesses:**

I think this paper is very clearly written and proposes a reasonable and fairly simple extension of previous shape reconstruction methods. I like that it makes a step towards neural implicit reconstructions that are more usable in practice (thank to the ability to accommodate higher levels of detail) while still retaining the benefits of a data driven approach. The experimental results are pretty convincing, and there is a clear advantage in the comparisons.

One drawback is the requirement to overfit a model (which takes ~30 min) for each new target set of parts. Additionally, the method is currently not always able to extrapolate local detail when parts are edited, since the overfit model is limited to the known regions of the parts. This detracts a bit from the claim of being able to make semantically reasonable edits at a high level of detail.

---

> ### Author Response · Authors · 2022-08-01
> **Rebuttal**
>
> Thanks for the feedback and suggestions. We will correct all smaller issues; below we address the main concerns:
>
> ### Long overfitting times
> We used a conservatively long overfitting schedule that was shared by all shapes in our experiment, to make sure the overfitting results for each shape had fully converged. Note that this overfitting already starts from the generalizable model and refines it. We can reduce this time without significant loss in accuracy, for example overfitting for only 5 minutes increases Chamfer distance and SDF error by only ~10% (0.017 to 0.019 CD and 0.679 to 0.744 SDF - these numbers are roughly comparable to Table S1 in the supplementary, but not fully comparable, since they were computed on a smaller subset than Table S1). Additionally, there has been active research aimed at reducing the overfitting time for neural representations (mostly for NeRFs), for example using hash tables and CUDA kernels [A] which takes order of a minute. Adapting this work to our setting seems like a fertile direction for future research that we hope can draw inspiration from our method.
>
> ### Detail near joint regions
> Good point. Since the joint regions are lower-detail than non-joint regions, details may stop before reaching the joint regions — as in the seat slate mentioned by the reviewer. Since we use bounding boxes to define joint regions, the estimated joint regions are conservative estimates of the actual joint regions, which may be smaller. An interesting future exploration would be to use a tighter definition of joint regions, for example by making use of the coarse geometry, or by learning the extent of joint regions or the falloff between joint and non-joint regions. We will add a discussion to Section 4.
>
> ### Applications in Shape Compression
> Thanks for the interesting suggestion, we have not experimented with shape compression, but we could imagine that a variant of ours, where the overfitted model is only used to represent regions that are not already well-represented by the prior, could improve compression by using smaller overfitted and generalizable networks. Using asymmetric network sizes might also be an interesting direction for future work. While it would require a mechanism for asymmetric weight blending, it might also allow the generalizable network to be larger, since it only needs to be stored only once per shape category (it can act like a predefined dictionary for the category). We will add a discussion in our future work passage.
>
> ### Reference
> [A] Instant Neural Graphics Primitives with a Multiresolution Hash Encoding, Müller et al., Siggraph 2022

---

### Meta-Review · Area_Chair_J1eD · 2022-08-24

**Recommendation:** Accept
**Confidence:** Certain

**Metareview:**

All the reviewers appreciated this work combining generalization capabilities of shape-space models and the fidelity achieved in a single shape overfit/optimization.

**Award:**

No

---

### Decision · Program_Chairs · 2022-09-14

Accept